# Are handcrafted filters helpful for attributing AI-generated images?

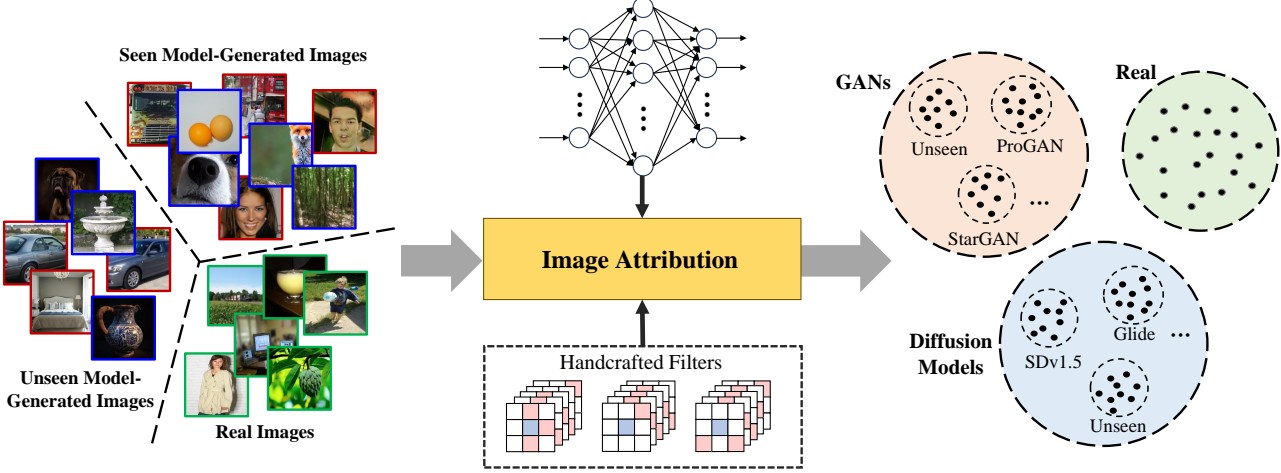

**Figure 1: The basic concept of our propose method. We explore the possibility to design and incorporate handcrafted filters to attribute an image to real, one of the GANs and DMs that are seen in training, unseen GANs, and unseen DMs.**

## ABSTRACT

Recently, a vast number of image generation models have been proposed, which raises concerns regarding the misuse of these artificial intelligence (AI) techniques for generating fake images. To attribute the AI-generated images, existing schemes usually design and train deep neural networks (DNNs) to learn the model fingerprints, which usually requires a large amount of data for effective learning. In this paper, we aim to answer the following two questions for AI-generated image attribution, 1) is it possible to design useful handcrafted filters to facilitate the fingerprint learning? and 2) how we could reduce the amount of training data after we incorporate the handcrafted filters? We first propose a set of Multi-Directional High-Pass Filters (MHFs) which are capable to extract the subtle fingerprints from various directions. Then, we propose a Directional Enhanced Feature Learning network (DEFL) to take both the MHFs and randomly-initialized filters into consideration. The output of the DEFL is fused with the semantic features to produce a compact fingerprint. To make the compact fingerprint discriminative among different models, we propose a Dual-Margin Contrastive (DMC) loss to tune our DEFL. Finally, we propose a reference based fingerprint classification scheme for image attribution.

Experimental results demonstrate that it is indeed helpful to use our MHFs for attributing the AI-generated images. The performance of our proposed method is significantly better than the state-of-the-art for both the closed-set and open-set image attribution, where only a small amount of images are required for training.

## CCS CONCEPTS

• **Computing methodologies** → **Computer vision**; **Artificial intelligence**.

## KEYWORDS

AI-Generated Image Attribution, Handcrafted Filters, Model Fingerprint

## 1 INTRODUCTION

Nowadays, a vast amount of artificial intelligence (AI) image generation technologies have been proposed [2, 7, 11, 14, 17–19, 24, 25, 30, 31, 33, 39, 43], which are shown to have a great potential in entertainment, education, and art design. Despite the advantage, these schemes may be maliciously used to generate fake images (termed as AI-generated images) that would cause negative impacts on the society, raising public concerns. For example, an AI-generated image of Pentagon explosion had deceived a lot of people, which was believed to be responsible for a sudden decline in the US stock market [27]. Therefore, it is urgent to develop effective schemes that are able to correctly identify the AI-generated images.

Most of the researchers focus on the task of AI-generated image detection, which tells whether an image is real or AI-generated [8, 15, 16, 21, 26, 35, 37, 38]. Wang *et al.* [37] attempt to detect images generated by generative adversarial networks (termed as

GAN-generated images), which is conducted by training a ResNet50 [13] for binary image classification. Later, the frequency cues [8, 15, 16], texture features [21], and gradient maps [35] are explored for GAN-generated image detection, which achieve promising results. Recently, with the remarkable success of diffusion models in image generation, researchers have started to work on the detection of images generated using diffusion models (termed as DM-generated images). The diffusion model based image reconstruction errors [38] and the features extracted from CLIP image encoder [29] are utilized for the detection of DM-generated images [26].

To hold the model owner responsible for model misuse, it is necessary to further identify which source model is used to generate the fake image. In literature, such a task is termed as AI-generated image attribution or model attribution. Similar to the detection task, a majority of the AI-generated image attribution schemes pay attention to the identification of generative adversarial networks (GANs) from the fake images. Researchers propose different types of fingerprints for attributing GAN-generated images [3, 8, 40, 42], which try to effectively capture the specific model information in the GAN-generated images. However, these schemes are limited to closed-set image attribution which require the source GANs to be seen during training. To deal with this issue, a few attempts have been made on open-set image attribution [9, 41], which is capable of identifying the images generated from unseen GANs. Sha *et al.* [32] take advantage of the prompts, which are used to generate the images, for attributing fake images generated by text-to-image models. This approach requires accurate estimation of prompts, which limits its application in real-world scenarios.

The main challenge of AI-generated image attribution is how we could learn fingerprints that are representative to different models. So far, researchers tend to conduct such a fingerprint learning by using deep neural networks (DNNs), which neglects the importance of handcrafted filters in capturing the traces left by image generation models. In addition, it may require to collect a large amount of training images for effective learning. Be aware of this, we try to answer in this paper how we could design good handcrafted filters to facilitate and fasten the fingerprint learning.

On the other hand, we notice that the development of AI image generation models is very fast. It may be not sufficient to consider image attribution for a single category of generation models. Instead of focusing on attributing GAN-generated images or DM-generated images as what have been done in the literature, we aim to do the image attribution by taking both GANs and DMs into consideration. As shown in Fig. 1, by taking advantage of the handcrafted filters and neural networks, our proposed method attributes an image to real, one of the GANs and DMs that are seen in training, unseen GANs, and unseen DMs.

Concretely, we design a set of handcrafted Multi-Directional High-Pass Filters (MHFs) that are able to capture diverse and subtle traces left by specific models from different directions. Then, we propose a Directional Enhanced Feature Learning network (DEFL) by incorporating both the MHFs and randomly-initialized convolutional filters. Given an image for input, the output of the DEFL is fused with the features extracted from a CLIP image encoder to obtain a compact fingerprint. To have a good differentiation among the real, GAN and DM-generated images, we propose a Dual-Margin Contrastive (DMC) loss to tune the DEFL. With the

compact fingerprint available, we further propose a reference based fingerprint classification to attribute the image to a proper class. The experiments demonstrate the usefulness of our handcrafted filters in attributing AI-generated images to seen and unseen GANs and DMs. With the help of the MHFs, our proposed method is shown to be significantly better than the state-of-the-art schemes. In addition, we only need a small amount of images for training. The main contributions are summarized below.

- We explore the effectiveness of handcrafted filters in attributing AI-generated images, where we design a set of Multi-Directional High-Pass Filters (MHFs) to extract subtle traces for specific models.
- We propose a Directional Enhanced Feature Learning network (DEFL) by taking both the handcrafted filters and randomly-initialized filters into consideration. The DEFL is tuned by a Dual-Margin Contrastive (DMC) loss that is newly designed.
- We propose a reference based fingerprint classification for attributing the AI-generated images to different source models.

## 2 RELATED WORK

### 2.1 AI Image Generation

There are two main categories of AI image generation schemes, including GAN-based image generation and [2, 10, 17–19, 24, 39, 43] and DM-based image generation [7, 11, 14, 25, 30, 31, 33].

GAN generates realistic images through adversarial training between the generator and discriminator. Karras *et al.* propose Pro-GAN [18] by progressive training, which is able to accelerate the training while ensuring the stability of high-resolution image generation. Karras *et al.* propose StyleGAN [19] with a style-based generator to control the visual representation layer by layer. Other popular GANs include CGAN [24], StackGAN [43], AttnGAN [39], BigGAN [2], and GigaGAN [17], which are proposed for high-quality text-to-image generation.

Compared to GANs, diffusion models (DMs) are easier to train and shown to be more promising in image generation. Ho *et al.* propose DDPM [14] to pioneer the research in DM-based image generation, which samples a random noise and then iteratively generates an image from the noise. DDIM [33] reduces the computational complexity of DDPM, which is able to generate the images with fewer sampling steps. Due to the advantages of diffusion models, tremendous efforts have been paid in the field of DM-based image generation, where numerous models are proposed to generate high-quality images, including ADM [7], LDM [31], Glide [25], DALLE2 [30] and VQDM [11].

### 2.2 AI-Generated Image Detection

AI-generated image detection aims to identify whether an image is real or AI-generated. Earlier works focus on detecting GAN-generated images. Wang *et al.* propose [37] to detect the GAN-generated images by training Resnet50 [13] for binary image classification. Frank *et al.* [8] observe consistent traces of GAN-generated images in the high frequency domain and train the detector by utilizing the high-frequency features. Jeong *et al.* further [16] train

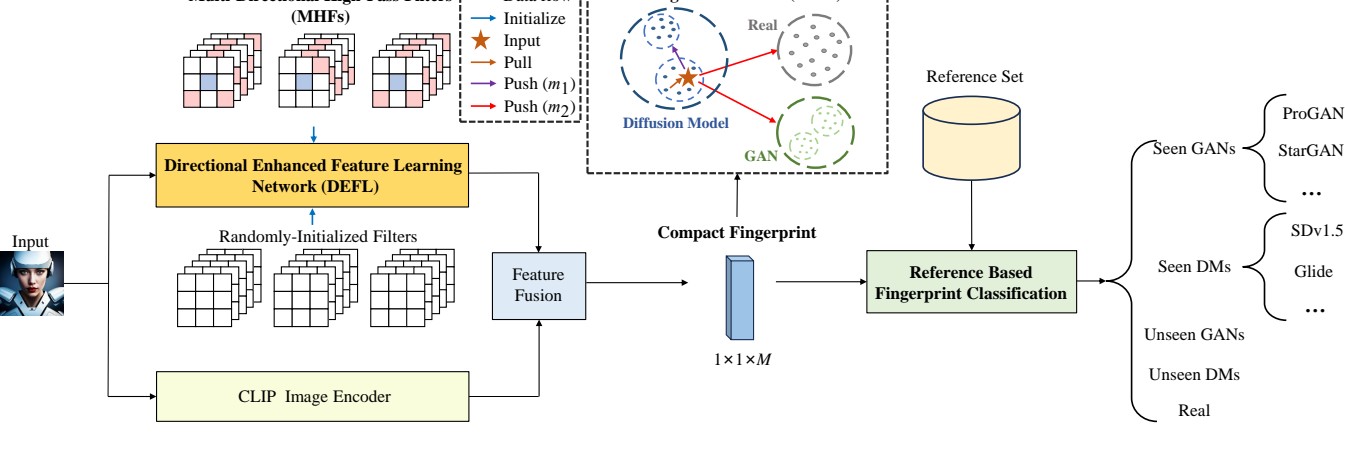

**Figure 2: An overview of our proposed method for AI-generated image attribution.**

the detector using frequency-disturbed images to improve the generalization ability. The works in [21] and [35] explore the usage of the texture features and gradient maps for GAN-generated image detection, which achieve relatively good performance.

Recently, researchers start to pay attention to DM-generated image detection. Wang *et al.* [38] utilize the diffusion model to reconstruct the input image, and then compute the difference between the input image and its reconstructed version for DM-generated image detection. Ojha *et al.* [26] leverage image features extracted by CLIP image encoder[29] to facilitate the detection, which is shown to be applicable for detecting the GAN or DM-generated images.

### 2.3 AI-Generated Image Attribution

AI-generated image attribution (also termed as model attribution) is to identify the source models of the AI-generated images, i.e., which model is used to generated the image. Yu *et al.* [42] validate the existence of fingerprints in GAN-generated images by using a deep convolutional neural network supervised by image-source pairs. Frank *et al.* [8] propose to extract the fingerprint from the artifacts in the DCT spectrum to attribute the image to different GANs. Bui *et al.* [3] propose a feature mixing mechanism to synthesize new data for attribution of GAN-generated images whose semantics and transformations are unseen in training. Yang *et al.* [40] incorporate contrastive learning among different image patches to extract the fingerprint that is representative to GANs with the same architecture but trained using different seeds, datasets and loss functions.

The aforementioned schemes assume all the GANs are seen in training, which work in a closed-set image attribution scenario. They are not applicable or do not work well for attributing the images generated from unseen models. To deal with this issue, Girish *et al.* [9] first consider the task of open-set image attribution, they cluster features from seen GANs and recognize unseen GANs by detecting features outside of clusters. Yang *et al.* [41] generate open-set samples to simulate the traces of unseen GANs, which are then used to train a network capable of image attribution for unseen GANs. Instead of simulating the unseen GANs, the work in [34] directly uses some samples from unseen GANs in advance

for training a open-set image attribution network. The work on attributing DM-generated images is very limited, Sha *et al.* [32] train the classifier by combining images and prompts.

Despite the progresses that have been made in AI-generated image attribution. It seems that the researchers have the tendency to completely hand over the task to neural networks. Meanwhile, most of the existing works consider to attribute the images to a single category of generation models (either GANs or DMs). In this paper, we explore the possibility and effectiveness of designing handcrafted filters for AI-generated image attribution, where we take both the seen and unseen GANs and DMs into consideration to meet the needs of the continuous development of image generation models.

## 3 PROPOSED METHOD

Fig.2 gives an overview of our proposed method. We first handcraft a set of Multi-Directional High-Pass Filters (MHFs) to effectively extract the subtle traces left by different models. The MHFs are combined with a set of randomly-initialized filters to construct a Directional Enhanced Feature Learning network (DEFL) to learn model-representative features. These features are then fused with the semantic features (extracted by a CLIP image encoder [29]) to obtain a compact fingerprint. The training of the DEFL is supervised by a Dual-Margin Contrastive (DMC) loss that is newly designed to make the compact fingerprint more discriminative for real, DM-generated and GAN-generated images. The compact fingerprint is eventually used for attributing the images, which is performed by a reference based fingerprint classification scheme.

### 3.1 Multi-Directional High-Pass Filters

We notice that the existing schemes tend to learn features representative to models by training a neural network, which may require a large amount of training images for good performance. For AI-generated images, most of the model fingerprints may be subtle, which mainly exist in the high-frequency component. A straightforward strategy to extract the subtle fingerprints is using some existing high-pass filters. However, the subtle fingerprints of different models may be very diverse and appear in various directions.

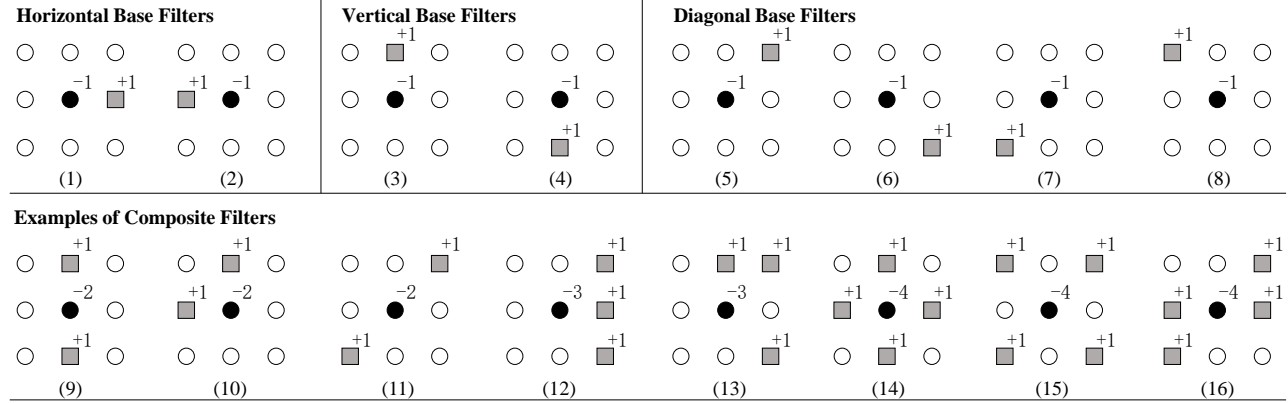

Figure 3: The Base filters and some examples of the composite filters.

Using existing high-pass filters is not sufficient to comprehensively capture these fingerprints. Here, we design a set of handcrafted Multi-Directional High-Pass Filters (MHFs) that are able to extract the subtle fingerprints from different directions.

Each of our MHFs is sized 3×3 with the center denoted as $h(x, y)$. The basic concepts of designing the MHFs are two folded, 1) the summation of the coefficients should be zero, which is consistent with ordinary high-pass filters, and 2) the polarity of coefficients along a certain direction should be opposite to capture the residual.

**Base Filters.** We first design a set of base filters for three directions including horizontal, vertical and diagonal base filters. For each horizontal base filter, we set the center coefficient as $h(x, y) = -1$ and one of its horizontal neighbors as 1 (i.e., $h(x - 1, y) = 1$ or $h(x + 1, y) = 1$), the rest of which are assigned as zero. In total, there are two horizontal base filters as shown in Fig. 3(1) and (2). Similarly, we set the center coefficient as $h(x, y) = -1$ and one of its vertical neighbors as 1 (i.e., $h(x, y - 1) = 1$ or $h(x, y + 1) = 1$) to form a vertical base filter. While a diagonal base filter is designed by setting the center coefficient as $h(x, y) = -1$ and one of its diagonal neighbors as 1 (i.e., $h(x - 1, y - 1) = 1$, $h(x - 1, y + 1) = 1$, $h(x + 1, y - 1) = 1$ or $h(x + 1, y + 1) = 1$). As such, we have two vertical base filters (see Fig. 3(3) and (4)) and four diagonal base filters (see Fig. 3(5)-(8)).

**Composite Filters.** To improve the diversity of the base filters and to sufficiently capture the subtle fingerprints, we further derive a set of composite filters according to the 8 base filters. In particular, we select $n \leq 8$ base filters to form a composite filter by

$$h_c = h_1 * h_2 ... * h_n, \quad (1)$$

where $*$ is the convolution operation and $h_1$, $h_2$, ... and $h_n$ refer to the selected based filters. In total, we conduct a set of 246 different composite filters. Fig. 3(9)-(16) illustrate some of the composite filters. Take Fig. 3(12) as an example, this composite filter is computed using the base filters given in Fig. 3(1), (5) and (6). The 8 base filters and 246 composite filters form the set of MHFs which contain 254 handcrafted filters.

## 3.2 Network Architecture of DEFL

Fig. 4 shows the network architecture of our DEFL, which progressively extracts model-representative features in four levels. In each level, the network is established by a Directional Convolutional Block (DCB) and a Standard Convolutional Block (SCB). The DCB and SCB share the same structure, both of which contain 64 3 × 3 filters for convolution followed by batch normalization (BN) and ReLU activation. The difference relies on the initialization of the filters. The DCB uses a set of 64 MHFs for initialization, which is responsible to extract model-representative features from the high-frequency component across various directions. The SCB randomly initializes the filters to extract other model-representative features for compensation. The outputs of the DCB and SCB are concatenated to form 128 feature maps which are fed into the next level.

We randomly partition our MHFs (254 filters in total) into four parts, where each of the first three parts contains 64 distinct filters and the last part contains 62 distinct filters. The filters in each part are served as the initial filters in each DCB. Note that we randomly duplicate two of the filters for the last part, such that the number of filters is 64. During the filter partition, we make sure that each part contains filters with the value of the center coefficients ranging from -1 to -7, which are the maximum and minimum of values of the center coefficients according to Eq. (1). As such, each part corresponds to a compact representation of the whole set of MHFs. By progressively using the DCBs and SCBs, the distinctiveness of the features gradually improves to capture a wide range of traces left by different models. On the other hand, since our MHFs are already capable of extracting the subtle fingerprints from various directions, we only need a small amount of data to fine tune the DEFL for optimal feature learning.

## 3.3 Semantic Feature Extraction and Fusion

Semantic features are shown to be a good complementary for learning model fingerprints [26, 32]. Here, we adopt the CLIP image encoder proposed in [29] to obtain the semantic features, which are then fused with the directional enhanced features extracted from our DEFL. We adopt the Resnet50 [13] as the feature fusion network, where the output of the final fully connected layer is considered as

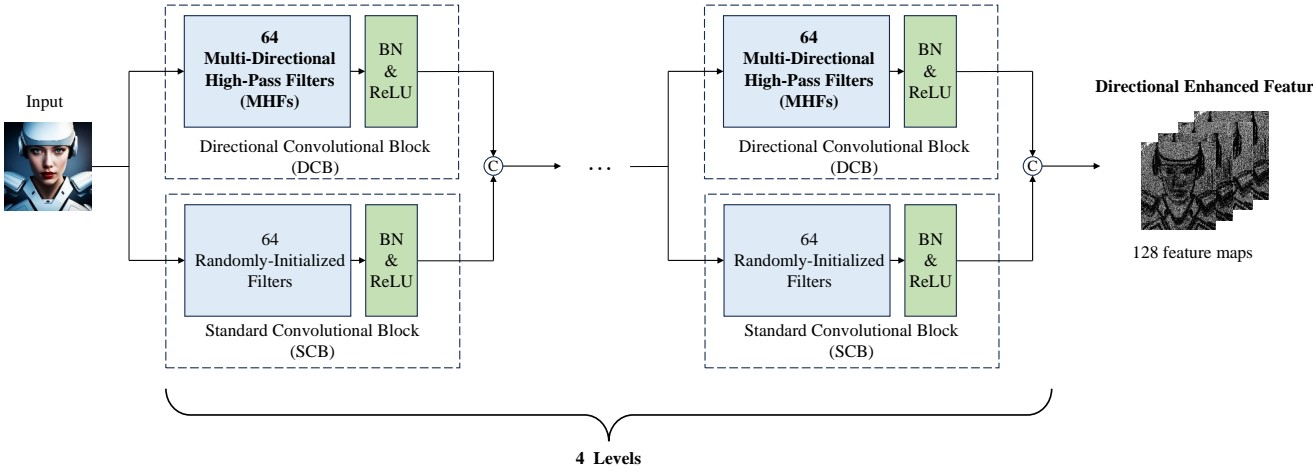

**Figure 4: The architecture of Directional Enhanced Feature Learning network (DEFL).**

the model fingerprint. This fingerprint is very compact, which is a 2048-dimensional vector.

## 3.4 Dual-Margin Contrastive Loss

Given a pair of two images $x_i$ and $x_j$, we want to train our DEFL such that the corresponding fingerprints (say $f_i$ and $f_j$) have the minimized distance if they are generated from the same model. When they are generated from two different models, we believe a representative fingerprint should have the following properties, 1) if the two models are both GANs or DMs, their fingerprints should not differ too much because the two models share similar architecture, and 2) if one model is GAN and the other is DM, their fingerprints should be far away due to the dissimilarity between the GANs and GMs. To this end, we propose in this section a Dual-Margin Contrastive (DMC) loss, where a small margin is adopted to fulfill the requirement of the first property and a larger margin is used for the second property.

Let's denote $y_{ij}$ as a label to indicate whether $x_i$ and $x_j$ are from the same class, where $y_{ij} = 1$ means both of them are real or generated from the same model, otherwise $y_{ij} = 0$. For the same token, we denote $z_{ij}$ as a label to tell whether the two images are generated using similar methods. We set $z_{ij} = 1$ if both of them are real, GAN-generated or DM-generated, otherwise $z_{ij} = 0$. Our DMC loss is formulated below.

$$
\begin{aligned}
\mathcal{L}_M = &\frac{1}{B^2} \sum_{i=1}^{B} \sum_{j=1}^{B} \Big\langle y_{ij} \cdot d(f_i, f_j) \\
&+ z_{ij} \cdot (1 - y_{ij}) \cdot max(0, m_1 - d(f_i, f_j)) \\
&+ (1 - z_{ij}) \cdot max(0, m_2 - d(f_i, f_j)) \Big\rangle,
\end{aligned}
\tag{2}
$$

where $B$ is the batch size, $m_1$ and $m_2$ are the margins with $m_1 < m_2$, $d(f_i, f_j)$ computes the Euclidean distance between $f_i$ and $f_j$, and $max(a, b)$ returns the maximum between $a$ and $b$. By using such a DMC loss, we push the fingerprints of a GAN-generated image and a DM-generated image to differ over a larger margin (i.e., $m_2$), to better distinguish between GANs and DMs. Meanwhile, we push the fingerprints of the GAN or DM-generated images to differ over

a small margin (i.e., $m_1$), which is helpful for accurate attribution within the GANs or DMs.

## 3.5 Reference Based Fingerprint Classification

In this section, we propose to classify our compact fingerprints based on a reference set which is a subset of the training data. Let's assume that there are in total $N$ different classes in the reference set, each of which contains $M$ images. We extract the compact fingerprints from the reference set to construct a set of reference fingerprints. We denote the $m$th reference fingerprint in class $n$ as $f_{mn}$, where $m \in [1, M]$ and $n \in [1, N]$.

Given a test image with its compact fingerprint $f$ extracted, we compute the average distance between $f$ and the reference fingerprints in class $n$ as

$$
d_n = \frac{1}{M} \sum_{m=1}^{M} d(f, f_{mn}).
\tag{3}
$$

As such, we obtain a vector $\mathbf{d} = \{d_1, d_2, ..., d_N\}$ to represent how close the test image is to different classes.

We denote the minimum element in $\mathbf{d}$ as $d_{min}$. If $d_{min}$ is less than or equal to a threshold $\theta$, we consider the compact fingerprint $f$ to be from the $k$th class where

$$
k = \underset{n}{argmin}\, d_n.
\tag{4}
$$

If $d_{min} > \theta$, $f$ will be classified as a fingerprint of an unseen model which is not used to generate the fake images in the reference set. To further determine whether the unseen model is a GAN or DM, we compute the center of the reference fingerprints of all the GAN-generated images, which is denoted as $c_g$ for simplicity. Similarly, we compute the center $c_d$ of the reference fingerprint of all the DM-generated images. The unseen model is eventually attributed as a GAN if

$$
d_{min} - c_g < d_{min} - c_d.
\tag{5}
$$

Otherwise, we attribute the unseen model as a DM.

# 4 EXPERIMENTS

## 4.1 Setup

**Table 1: Description of the dataset used in the experiments.**

| Category | Model | Seen | Train (Ref.) Number | Test Number |
|---|---|---|---|---|
| GAN | ProGAN [18] | Yes | 250 (50) | 2000 |
| | BigGAN [2] | Yes | 250 (50) | 2000 |
| | StarGAN [4] | Yes | 250 (50) | 2000 |
| | CycleGAN [44] | Yes | 250 (50) | 2000 |
| | StyleGAN [19] | No | - | 2000 |
| | GauGAN [28] | No | - | 2000 |
| DM | Glide [25] | Yes | 250 (50) | 2000 |
| | ADM [7] | Yes | 250 (50) | 2000 |
| | SDv1.5 [1] | Yes | 250 (50) | 2000 |
| | VQDM [11] | Yes | 250 (50) | 2000 |
| | DALLE2 [30] | No | - | 2000 |
| | Midjourney [23] | No | - | 2000 |
| Real | ImageNet [5] | Yes | 250 (50) | 2000 |

**Datasets.** In order to comprehensively evaluate the performance of our proposed method, we collect a dataset which contains real images and fake images generated from 12 different models, including 6 GANs and 6 DMs. The details of the dataset are listed in Table 1. The GAN-generated images are contributed by the work in [37], the DM-generated images are collected by GenImage [45], and the real images are selected from ImageNet [5]. For simplicity, we consider the real images to be generated from a seen model in the following discussions.

Unlike the state-of-the-art (SOTA) schemes which use thousands of images per model for training [3, 41], we only use 250 images generated from each seen model for training. While the number of the testing images per model is set to 2000, which is much larger than that of the training images. We assign 4 GANs and 4 DMs as seen models, while the remaining 2 GANs and 2 DMs are treated as unseen models with no images for training. For each seen model, we randomly select 50 images from the 250 training images to construct the reference set for fingerprint classification.

**Implementation details.** We set $m_1 = 5$, $m_2 = 10$ to compute our DMC loss, respectively. The threshold $\theta$ is set to 3.5 for reference based fingerprint classification. The DEFL and the fusion network (i.e., Resnet50) are tuned using the Adam optimizer [20] with a learning rate of $10^{-3}$. We use a pre-trained CLIP:RN50x16 image encoder [29] to extract the semantic features, the parameters of which are frozen during training.

**Methods for comparison.** We compare our method with eight SOTA schemes, including six schemes for closed-set image attribution (i.e., PRNUF [22], CNNF [42], DCT-CNN [8], RepMix [3], DNA-Det [40] and CPL [34]) and two open-set image attribution schemes (i.e., OPGAN [9] and POSE [41]). For fair comparisons, we use the training and testing set listed in Table 1 for all the methods.

**Evaluation scenarios.** We consider six scenarios for evaluation based on the dataset constructed according to Table 1, which are listed below:

- **Real/GAN/DM:** We attribute an image to real, specific GAN-generated or specific DM-generated. There are in total 9 classes for seen models, where the whole dataset is used for training and testing.
- **Real/GAN:** We attribute an image to real or specific GAN-generated. There are in total 5 classes for seen models, where the real and fake images generated from seen GANs are used for training and testing.
- **Real/DM:** We attribute an image to real or specific DM-generated. There are in total 5 classes for seen models, where the real and fake images generated from seen DMs are used for training and testing.
- **GAN/DM:** We attribute a fake image to specific GAN-generated or specific DM-generated. There are in total 8 classes for seen models, where all the fake images are used for training and testing.
- **GAN only:** We attribute a GAN-generated image to a specific GAN. There are in total 4 classes for seen models, where the fake images generated from seen GANs are used for training and testing.
- **DM only:** We attribute a DM-generated image to a specific DM. There are in total 4 classes for seen models, where the fake images generated from seen DMs are used for training and testing.

**Evaluation metrics.** For closed-set image attribution, we use the classification accuracy (termed as Acc) for the seen models as an indicator. For open-set image attribution, we follow the work in [41] to adopt the following four metrics, including Area Under ROC Curve (AUC), Open Set Classification Rate (OSCR) [6], Normalized Mutual Information (NMI), and Adjusted Rand Index (ARI). AUC is used to evaluate the performance of distinguishing images from seen or unseen models, OSCR is used to evaluate the classification accuracy of both the seen models and unseen models, NMI calculates the normalized mutual information between prediction and ground-truth, and ARI normalizes the accuracy of correctly grouped sample pairs. Since our method has the capability to identify the unseen models into unseen GANs or unseen DMs, we further calculate the accuracy of classifying an unseen model into GAN or DM for evaluation, which is termed as $Acc_u$ in the following discussions.

## 4.2 Performance Comparison

**Closed-set image attribution.** We evaluate the performance of each method in all the six scenarios, where the testing set only contains images from seen models. As shown in Table 2, our proposed method significantly outperforms the SOTA schemes, which achieves the best performance in all the six scenarios. Specifically, in the most challenging scenario "Real/GAN/DM" and "GAN/DM", the classification accuracy of our proposed method is 98.6% and 99.1%, which is over 30.4% and 31.1% higher than that of the SOTA schemes, respectively. The poor performance of the SOTA methods is probably due to the fact that they require a large amount of data for training, which are not appropriate to be used when the training data is limited.

**Open-set image attribution.** We evaluate the performance of our proposed method and the SOTA schemes which are dedicated

**Table 2: Performance comparisons among different schemes for closed-set image attribution.**

| Method | Real/GAN/DM (9 classes) | Real/GAN (5 classes) | Real/ DM (5 classes) | GAN/DM (8 classes) | GAN only (4 classes) | DM only (4 classes) |
|---|---|---|---|---|---|---|
| | $Acc(\%)$ | $Acc(\%)$ | $Acc(\%)$ | $Acc(\%)$ | $Acc(\%)$ | $Acc(\%)$ |
| PRNUF [22] | 36.0 | 47.6 | 41.2 | 39.3 | 59.0 | 48.0 |
| CNNF [42] | 68.2 | 76.0 | 81.6 | 68.0 | 83.5 | 79.5 |
| DCT-CNN [8] | 40.8 | 42.4 | 52.1 | 48.5 | 52.0 | 64.5 |
| DNA-Det [40] | 24.4 | 37.2 | 33.6 | 38.8 | 53.9 | 41.0 |
| RepMix [3] | 47.1 | 54.5 | 49.3 | 48.6 | 60.9 | 50.1 |
| OPGAN [9] | 54.8 | 77.6 | 70.4 | 57.5 | 88.0 | 85.5 |
| POSE [41] | 46.2 | 53.4 | 50.9 | 51.4 | 61.3 | 53.7 |
| CPL [34] | 33.3 | 44.4 | 55.6 | 42.0 | 55.0 | 65.5 |
| **Ours** | **98.6** | **99.6** | **99.2** | **99.1** | **99.5** | **99.5** |

**Table 3: Performance comparisons among different schemes for open-set image attribution.**

| Method | Real/GAN/DM (9 seen classes, 4 unseen classes) | | | | | GAN/DM (8 seen classes, 4 unseen classes) | | | | |
|---|---|---|---|---|---|---|---|---|---|---|
| | $AUC(\%)$ | $OSCR(\%)$ | $NMI$ | $ARI$ | $Acc_u(\%)$ | $AUC(\%)$ | $OSCR(\%)$ | $NMI$ | $ARI$ | $Acc_u(\%)$ |
| OPGAN [9] | 52.4 | 37.2 | 0.20 | 0.14 | - | 53.1 | 39.7 | 0.20 | 0.16 | - |
| POSE [41] | 50.3 | 33.2 | 0.13 | 0.10 | - | 50.6 | 35.8 | 0.15 | 0.11 | - |
| **Ours** | **90.8** | **90.3** | **0.33** | **0.28** | **74.5** | **92.0** | **91.6** | **0.37** | **0.35** | **78.9** |

for open-sent image attribution, including OPGAN [9] and POSE [41]. We consider the scenarios of "Real/GAN/DM" and "GAN/DM" for evaluation, where the testing set contains images from both the seen models and unseen models. It can be seen from Table 3 that, in both scenarios, our proposed method is significantly better than the SOTA schemes regardless of the performance indicators. Both the *AUC* and the *OSCR* of our scheme are over 90% in two scenarios, which are over 40% higher than the SOTA schemes. Moreover, our proposed method can effectively attribute unseen models to unseen GANs or unseen DMs, where the $Acc_u$ is 74.5% and 78.9% in the scenario "Real/GAN/DM" and "GAN/DM", respectively. It should be noted that the existing schemes are unable to further classify unseen models, so no $Acc_u$ is reported for them.

**t-SNE visualization.** We visualize the distribution of compact fingerprints by t-SNE [36], as shown in Fig. 5. It can be seen that, in closed-set image attribution, the fingerprints from the same models are tightly clustered, where sufficient margins are maintained among different clusters. In open-set image attribution, the fingerprints of seen models have a similar clustering pattern as that of the closed-set attribution. For unseen models, their fingerprints would not cluster together with those from any seen models. We also observe that the fingerprints from unseen GANs are closer to those from seen GANs, while those from unseen DMs tend to appear around those from seen DMs. This indicates the ability of our compact fingerprints in differentiating the unseen GANs from the unseen DMs.

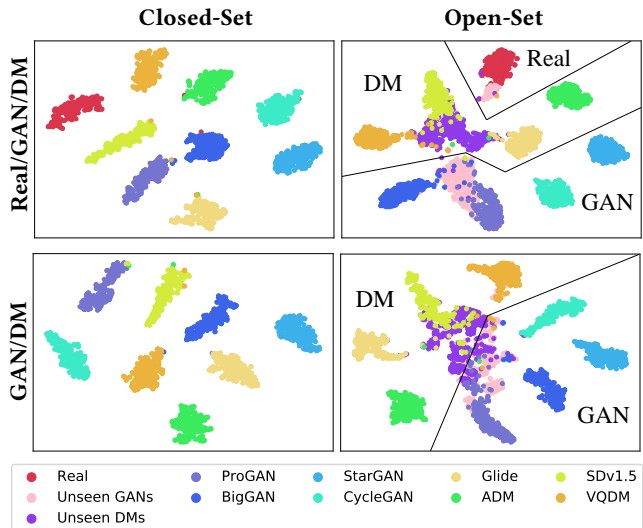

**Figure 5: The t-SNE visualization in the scenario "Real/GAN/DM" and "GAN/DM" for closed-set and open-set image attribution.**

## 4.3 Robustness

In real-world applications, images are spread across different social media platforms. When uploading the images into the social network platforms, it is likely that the images are further operated

**Table 4: The robustness of different schemes in the scenario "GAN/DM" for closed-set image attribution.**

| Method | Original Acc(%) | JPEG Acc(%) | Downsampling Acc(%) |
|---|---|---|---|
| PRNUF [22] | 39.3 | 38.3 | 11.2 |
| CNNF [42] | 68.0 | 63.8 | 16.6 |
| DCT-CNN [8] | 48.5 | 47.0 | 12.5 |
| DNA-Det [40] | 38.8 | 12.7 | 12.3 |
| RepMix [3] | 48.6 | 47.6 | 34.2 |
| OPGAN [9] | 57.5 | 50.1 | 36.9 |
| POSE [41] | 51.4 | 46.6 | 37.8 |
| CPL [34] | 42.0 | 40.9 | 31.1 |
| **Ours** | **99.1** | **90.5** | **43.8** |

**Table 5: Ablation studies by switching off each component in the scenario "GAN/DM" for closed-set and open-set image attribution.**

| Switch Off | Closed-Set Acc(%) | Open-Set AUC(%) | OSCR(%) | $Acc_u$(%) |
|---|---|---|---|---|
| DEFL | 77.2 | 84.8 | 72.3 | 78.8 |
| MHFs | 93.3 | 83.5 | 66.4 | 72.6 |
| DMC loss | 98.9 | 68.1 | 67.4 | 72.1 |
| RFC | 93.4 | - | - | - |
| - | **99.1** | **92.0** | **91.6** | **78.9** |

by the servers. In this section, we evaluate the robustness of our proposed method as well as the existing schemes in resisting the image operations that may be performed by the social network servers. We notice that the servers of social network platforms tend to compress or resize the images for the sake of reducing storage and bandwidth. Therefore, we consider two image operations here, the JPEG compression and downsampling, to evaluate the robustness. Table 4 gives the performance of different schemes after the JPEG compression (with a quality factor of 95) and downsampling (with a rescaling factor of 1/4). It can be seen that there is a noticeable performance degradation of all the methods. However, our proposed method still outperforms all the other schemes in resisting both image operations.

### 4.4 Ablation Studies

In this section, we verify the effectiveness of each component in our proposed method, including the Directional Enhanced Feature Learning network (DEFL), Multi-Directional High-Pass Filters (MHFs), Dual-Margin Contrastive (DMC) loss, and reference based fingerprint classification (termed as RFC for short). For each of the ablation studies, we switch off a component or replace a component with an existing scheme. Then, we rerun the experiments in the scenario "GAN/DM" to see how the performance is affected. Table 5 shows the results of ablation studies for closed-set and open-set

image attribution. Next, we elaborate the four ablation studies in detail.

**Effectiveness of the DEFL.** We switch off our DEFL and generate the compact fingerprints by only using the semantic features extracted by the CLIP image encoder [29]. It can be seen from Table 5 that the performance severely drops, especially for the task of closed-set image attribution, where the Acc is over 21.9% lower compared with using our proposed DEFL. This indicates the ability of our DEFL to learn model-representative features for the generation of discriminative fingerprints.

**Effectiveness of the MHFs.** We randomly initialize the filters in the DCBs in DEFL instead of using our proposed MHFs. In such a case, the Acc of the closed-set image attribution is 93.3% and the AUC of the open-set image attribution is 85.5%, which are 5.8% and 8.5% lower than using the MHFs for initialization, respectively (see Table 5). Therefore, our MHFs are indeed helpful for the task of image attribution.

**Effectiveness of the DMC loss.** We replace our DMC loss with a conventional contrastive loss which employs a single margin [12]. We can see from Table 5 that the performance of the closed-set image attribution slightly declines compared with using our DMC loss. However, when it comes to the open-set image attribution, the performance severely drops with the AUC of 68.1%, which is 23.9% lower than that of using our DMC loss. This demonstrates the advantage of our DMC loss, especially for the task of open-set image attribution.

**Effectiveness of the RFC.** We replace our RFC with a multi-class classification head to be trained together with the compact fingerprint extraction network, where the cross-entropy loss is used for training. In such a case, we are only able to conduct the closed-set image attribution, and the corresponding Acc is 93.4%, which is 5.7% lower than using the RFC. Therefore, our RFC is important for both the closed-set and open-set image attribution.

## 5 CONCLUSION

In this paper, we explore the possibility of designing and using handcrafted filters for AI-generated image attribution. To this end, we design a set of handcrafted Multi-Directional High-Pass Filters (MHFs) to extract subtle and representative model fingerprints from different directions. The MHFs are combined with a set of randomly-initialized filters to construct a Directional Enhanced Feature Learning network (DEFL). The output of DEFL is fused with the semantic features to generate a compact fingerprint. To obtain discriminative compact fingerprints for real, GAN-generated, and DM-generated images, we propose a Dual-Margin Contrastive loss to train the DEFL. Based on the compact fingerprints, we propose a reference based fingerprint classification for image attribution. Extensive experiments demonstrate the advantage of our proposed method over the SOTA schemes in various scenarios for both closed-set and open-set image attribution, where only a small amount of images are required for training.

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
