# OpenReview forum: "Are handcrafted filters helpful for attributing AI-generated images?"
_acmmm.org/ACMMM/2024/Conference — MM2024 Poster_

### Official Review · Reviewer_gYix · 2024-05-12

**Rating:** 3
**Confidence:** 3

**Summary:**

This work aims to attribute an AI-generated image by incorporating handcrafted filters. The authors propose multi-directional high-pass filters to extract subtle fingerprints. They also propose a directional enhanced feature learning network, which is tuned by a designed dual-margin contrastive loss, for AI-generated image attribution. A reference-based fingerprint classification is proposed to attribute the AI-generated image to different source models.

**Strengths:**

This paper proposes to use handcrafted filters combined with deep features to extract fingerprints for AI-generated image attribution. It is interesting to combine handcrafted features in this task. The authors also propose a loss function to improve the performance, where similarities between different models/classes are considered. The experiments are comprehensive, and the results look great compared with other baselines for both closed-set and open-set image attributions. The overall writing is easy to follow.

**Limitations:**

Despite the contributions of this paper, more clarification and details are needed for the proposed components. The motivations for the proposed architecture are lack of explanation. The proposed loss function seems to be a simple application from previous work. Some experiments are missing so that hard to justify the effectiveness of the proposed components.
1.	What are the models' representative features? Why does the proposed MHFs is effective to extract subtle traces generated by different models?
2.	Why use CLIP to extract visual features? For both GANs and DMs?
3.	For the existing high-pass filters, what do you refer to? What is the novelty of the proposed MHFs?
4.	For the proposed MHFs, based on equation 1, there should be 255 filters in total, which filter is not used and why? Explanation needs to be added.
5.	Why do you need SCB in DEFL as the fusion network uses ResNet?
6.	The proposed DMC shares the similarity with double-margin contrastive loss [1], [2]. Is DMC a simple application of double-margin contrastive loss?
[1] Gómez-Silva, M. J., Armingol, J. M., & de la Escalera, A. (2017, February). Deep part features learning by a normalised double-margin-based contrastive loss function for person re-identification. In International Conference on Computer Vision Theory and Applications (Vol. 7, pp. 277-285). SCITEPRESS.
[2] Hao, J., Dong, J., Wang, W., & Tan, T. (2018, August). DeepFirearm: Learning discriminative feature representation for fine-grained firearm retrieval. In 2018 24th International Conference on Pattern Recognition (ICPR) (pp. 3335-3340). IEEE.
7.	The ablation study for w/ and w/o CLIP encoder and fusion module are missing. Hard to tell if the performance is attributable to a strong encoder or fusion module.



Other problems:
Limitations should be discussed in this paper.
Will the collected datasets or protocols be made publicly available for evaluation?

**Suitability:**

2

---

### Official Review · Reviewer_ozh1 · 2024-05-18

**Rating:** 4
**Confidence:** 2

**Summary:**

The paper focuses on the attribution of AI-generated images, particularly addressing concerns related to the potential misuse of image generation models. It proposes the use of Multi-Directional High-Pass Filters (MHFs) and a Directional Enhanced Feature Learning network (DEFL) to extract subtle fingerprints from various directions and enhance feature learning. The paper also introduces a Dual-Margin Contrastive (DMC) loss for tuning the DEFL and a reference-based fingerprint classification scheme for image attribution. The goal is to design handcrafted filters to facilitate fingerprint learning and reduce the amount of training data required for the effective attribution of AI-generated images.

**Strengths:**

The paper's strengths lie in its innovative use of handcrafted filters, the proposed approach's theoretical soundness, the thorough evaluation and comparison with existing methods, and the clarity of presentation, which enhances its applicability to real-world image attribution challenges.

**Limitations:**

- Why is the high-pass filter chosen over other filters? Please justify your choice or include a reference to support it.
- It appears that the MHF is used as the initial parameter. How can it be ensured that high pass filtering characteristics remain after being trained?
- Are ResNet and DEFL trained end-to-end or separately?
- What are the constraints associated with this approach?

**Suitability:**

2

---

### Official Review · Reviewer_Ew6m · 2024-05-20

**Rating:** 4
**Confidence:** 4

**Summary:**

This paper presents a method for attributing AI-generated images. Compared with existing schemes that design and train deep neural networks to learn model fingerprints, the authors propose a Directional Enhanced Feature Learning network (DEEL) to extract model fingerprints from different directions. This network incorporates both Multi-Directional High-Pass Filters (MHFs) and randomly-initialized convolutional filters. The DEEL facilitates faster fingerprint learning. Additionally, this paper considers both GANs and DMs for image attribution. To better distinguish GAN and DM-generated images, the authors propose a Dual-Margin Contrastive (DMC) loss to tune the DEEL.  Furthermore, to attribute AI-generated images to different source models, a reference-based fingerprint is introduced.

**Strengths:**

- This paper addresses the question of whether handcrafted filters are helpful for attributing AI-generated images. The author proposes a novel method to verify the effectiveness of handcrafted filters in capturing the traces left by image generation models.
- The author uses designed handcrafted filters to promote and accelerate fingerprint learning, reducing the number of training images required.
- The author considers the rapid development of AI image generation models and aims to achieve image attribution by considering both GANs and diffusion models (DM).
- The proposed method has outperformed state-of-the-art (SOTA) schemes in various scenarios for both closed-set and open-set image attribution.

**Limitations:**

1. Clarity and Workflow:
The clarity of the paper needs to be strengthened. The overall workflow of the proposed method is not specifically described, making it difficult for readers to understand how the individual components fit into the big picture. Additionally, the description of the Dual-Margin Contrastive Loss in Section 3.4 is not clearly enough explained, which can easily cause misunderstandings among readers (see specific points in the justification below).

2. Limited Training Data and Overfitting:
In the experimental section, the author only used a limited amount of training data. This may lead to overfitting, making it impossible for the model to generalize to new data effectively.

3. Choice of Composite Filters:
In Section 3.1, the author conducts a set of 246 different composite filters. The reason for choosing 246 filters is not explained. Providing a rationale for this specific number would help clarify the methodology.

4. Small Amount of Data for MHFs:
In Section 3.2, the paper states, “since our MHFs are already capable of extracting…”, leading to the conclusion that only a small amount of data is needed. This conclusion lacks experimental proof, and it would benefit from additional empirical validation.

5. Reference Set and Generalization Ability:
In Section 3.5, selecting a subset of the training data as a reference set is deemed unreasonable as it can affect the generalization ability of the model. Additionally, on line 573, the calculation formulas for “c_g” and “c_d” are missing, which hinders the reproducibility of the results.

6. Evaluation Scenarios in Open-set Image Attribution:
In Section 4.2, the author only considers the scenarios of “Real/GAN/DM” and “GAN/DM” for evaluation in the attribution of the open-set image. The rationale for this selection is not provided. Including a justification or exploring additional scenarios would strengthen the evaluation's comprehensiveness.


justification:

1.   In section 3.4, the symbols related to Eq.(2) i.e., yij ，zij  need to be explained clearly and I don’t quite understand the expression of
Eq.(2).
2.   For the overfitting problem of the model, it is necessary to add regularization items to the loss function.
3.   For the annotation in Figure 2,  the author can clearly describe the workflow of the whole method, which is conducive to
the readers’ understanding.
4.   In the experimental part, the author may need to verify the generalization ability of the model with more experiments.

Minor:
In line 498, the “GMs” should be the ”DMs” I think.

**Suitability:**

3

---

### Meta-Review · Area_Chair_TxNt · 2024-07-03

**Recommendation:** Accept (Poster)
**Confidence:** 5

**Metareview:**

The paper addresses concerns related to the potential misuse of image generation models.
The proposed research direction is very promising and all reviewers are inclined towards acceptance.